# Xylooligosaccharides Enhance Lettuce Root Morphogenesis and Growth Dynamics

**DOI:** 10.3390/plants13121699

**Published:** 2024-06-19

**Authors:** Meng Kong, Jiuxing He, Juan Wang, Min Gong, Qiuyan Huo, Wenbo Bai, Jiqing Song, Jianbin Song, Wei Han, Guohua Lv

**Affiliations:** 1Institute of Environment and Sustainable Development in Agriculture, Chinese Academy of Agricultural Sciences, Beijing 100081, China; hkhkmeng@163.com (M.K.); hikerjx@163.com (J.H.); lxnync1230@163.com (J.W.); huoqiuyan 2022@163.com (Q.H.); baiwenbo@caas.cn (W.B.); songjiqing@caas.cn (J.S.); 2Station of Dawenliu, Shandong Yellow River Delta Nature Reserve, Dongying 257509, China; 3Shandong Agri-tech Extension Center, Jinan 250013, China

**Keywords:** root growth, xylooligosaccharides, transcriptional, lettuce

## Abstract

Enhancing root development is pivotal for boosting crop yield and augmenting stress resilience. In this study, we explored the regulatory effects of xylooligosaccharides (XOSs) on lettuce root growth, comparing their impact with that of indole-3-butyric acid potassium salt (IBAP). Treatment with XOS led to a substantial increase in root dry weight (30.77%), total root length (29.40%), volume (21.58%), and surface area (25.44%) compared to the water-treated control. These enhancements were on par with those induced by IBAP. Comprehensive phytohormone profiling disclosed marked increases in indole-3-acetic acid (IAA), zeatin riboside (ZR), methyl jasmonate (JA-ME), and brassinosteroids (BRs) following XOS application. Through RNA sequencing, we identified 3807 differentially expressed genes (DEGs) in the roots of XOS-treated plants, which were significantly enriched in pathways associated with manganese ion homeostasis, microtubule motor activity, and carbohydrate metabolism. Intriguingly, approximately 62.7% of the DEGs responsive to XOS also responded to IBAP, underscoring common regulatory mechanisms. However, XOS uniquely influenced genes related to cutin, suberine, and wax biosynthesis, as well as plant hormone signal transduction, hinting at novel mechanisms of stress tolerance. Prominent up-regulation of genes encoding beta-glucosidase and beta-fructofuranosidase highlights enhanced carbohydrate metabolism as a key driver of XOS-induced root enhancement. Collectively, these results position XOS as a promising, sustainable option for agricultural biostimulation.

## 1. Introduction

The root system is an indispensable organ that anchors and supports plants while absorbing the essential water and nutrients necessary for their growth and development. Root morphology is highly plastic and responsive to environmental conditions, allowing plants to adapt swiftly to sudden changes [1]. This developmental plasticity is facilitated by complex sensing and signalling mechanisms that integrate both internal and external signals, enabling plants to modulate their root architecture in response to diverse environmental stimuli [2,3]. Understanding and enhancing root function is vital for improving crop stress resistance and promoting sustainable agricultural practices.

In recent years, oligosaccharides derived from plants or animals have gained attention as promising bioregulatory agents in ecological agriculture [4,5,6,7]. These compounds are safe, soluble, non-toxic, and environmentally friendly, making them ideal for sustainable agriculture [8]. They have been shown to enhance the efficacy of pesticides and fertilisers and to induce stress resistance in crops, offering a potential solution for increasing crop yields while reducing chemical inputs [9,10,11]. Among the various oligosaccharides, chitosan oligosaccharides (CSOSs), alginate oligosaccharides (AOSs), and xylooligosaccharides (XOSs) have demonstrated significant potential in regulating plant growth and productivity [8,12,13]. CSOSs, derived from crustaceans, are composed of 2–10 glucosamine units linked by β-1,4-glycosidic linkages [8]. AOSs, obtained from alginate, contain 2–20 monosaccharides, including β-D-mannuronic acid and α-L-guluronic acid, also linked by β-1,4-glycosidic bonds [14]. XOSs, the focus of this study, are derived from lignocellulosic materials like wheat straw, consisting of 2–7 D-xylose molecules linked by β-1,4-glycosidic bonds [12].

Recent studies have shown that foliar application of extremely low concentrations of exogenous oligosaccharides can significantly promote root growth [15,16,17]. For instance, applying AOS at concentrations of 10~50 mg L^−1^ to Chinese cabbage (*Brassica campestris* L.) markedly increased total root length, volume, and surface area [18]. Additionally, the expression of auxin (IAA) and brassinosteroids (BRs) was significantly enhanced in tomato roots treated with chitosan oligosaccharides, cellooligosaccharides, and xylooligosaccharides, respectively, indicating that these compounds influence root growth through phytohormone pathways [19,20]. The auxin signalling pathway is widely recognised for its role in root development. Evidence suggests that AOS may regulate rice root growth by modulating the expression of auxin-related genes (*OSYUCCA1*, *OSYUCCA5*, *OSIAA11*, and *OSPIN1*), thereby increasing IAA concentration [18]. Other studies have shown that galactoglucomannan oligosaccharides (GGMOs) can stimulate root growth through changes in auxin transport, potentially mediated by flavonoids [21]. Additionally, AOS can induce the release of Ca^2+^ from intracellular and extracellular sources, activating nitrogen metabolism in flowering Chinese cabbage. This suggests that calcium channels are involved in oligosaccharide-mediated root growth [21,22,23]. AOS-induced nitric oxide (NO) generation has also been identified as a novel mechanism regulating root growth, as AOSs up-regulate the gene expression and enzyme activity of nitrate reductase (NR) in wheat roots [24,25]. Most research on xylooligosaccharides has focused on their effects on aboveground plant growth, such as promoting photosynthesis and leaf growth [20,26]. Studies on root growth have been limited and primarily conducted on model plants like *Arabidopsis thaliana* [27,28]. Given the potential variability in the mechanisms of action of XOSs across different plant species, crop-specific studies are needed. Furthermore, the differences in the pathways through by which plant hormones and oligosaccharides such as XOS regulate root development remain largely unexplored.

Lettuce (*Lactuca sativa*), a widely cultivated leafy vegetable, is a key focus of agricultural research due to its nutritional value and market demand. Root development is crucial for the lettuce growth of lettuce and affects water and nutrient uptake [29]. In this study, we used RNA-seq technology to investigate the effects of XOS on gene expression in lettuce roots, with IBAP serving as a comparative control to elucidate the differences and similarities in their regulatory mechanisms. By revealing the complex network of metabolic and signalling pathways involved in XOS- and IBAP-mediated root growth regulation, this work provides valuable insights for the application of these substances in agriculture.

## 2. Results

### 2.1. Xylooligosaccharides Promote Lettuce Root Growth

To investigate the effects of xylooligosaccharides (XOSs) on lettuce root development, we quantified root dry weights and analysed root morphology two days after the final treatment. Additionally, we treated lettuce plants with indole-3-butyric acid potassium salt (IBAP), a well-established plant hormone, to compare the effects of exogenous XOS with those of a known hormone on root growth. As illustrated in Figure 1A,B, XOS treatment significantly increased root dry weight by 30.77% compared to the water-treated control (WT). No significant differences were observed when compared to the IBAP treatment. Root morphology analysis revealed that XOS application markedly enhanced total root length, volume, and surface area by 29.40%, 21.58%, and 25.44%, respectively, compared to WT (Table 1). Additionally, XOS treatment significantly stimulated the elongation of both adventitious and lateral roots (Figure 2A). Foliar application of XOS increased the length of adventitious and lateral roots by 28.55% and 29.86%, respectively, compared to WT. The ratio of adventitious to lateral root lengths remained constant (Figure 2B), indicating that XOS primarily promotes root growth by facilitating the elongation of adventitious roots. No significant morphological differences were detected between the XOS and IBAP treatments. Notably, shoot dry weight also increased significantly by 26.73% under XOS treatment compared to WT (Figure 3). This increase was not significantly different from the IBAP treatment, suggesting a similar overall enhancement of biomass. These findings collectively demonstrate that XOS effectively promotes the biomass and elongation of adventitious and lateral roots in lettuce, paralleling the effects observed with the exogenous application of IBAP [30].

### 2.2. Effect of Xylooligosaccharides on Phytohormones of Lettuce

To further elucidate the drivers underlying the observed root growth enhancement by exogenous xylooligosaccharides (XOSs), we analysed the concentrations of major phytohormones in lettuce roots. Our analysis revealed that XOS treatment significantly elevated the levels of key phytohormones compared to the WT (Figure 4). Specifically, the concentrations of indole-3-acetic acid (IAA), zeatin riboside (ZR), methyl jasmonate (JA-ME), and brassinosteroids (BRs) were increased by 30.02%, 50.90%, 33.77%, and 26.86%, respectively. These findings suggest a strong correlation between enhanced root growth and the up-regulation of endogenous phytohormones induced by XOS.

For comparative purposes, we also analysed the phytohormone levels in lettuce roots treated with indole-3-butyric acid potassium salt (IBAP). The results showed that IBAP treatment further elevated the concentrations of IAA, ZR, JA-ME, and BR, with increases of 34.80%, 90.06%, 14.30%, and 28.15%, respectively, compared to WT. Although IBAP treatment resulted in higher phytohormone levels than XOS, the overall trends in hormone modulation were consistent. Collectively, these results highlight that XOS significantly enhances root development in lettuce through the modulation of endogenous phytohormone levels. The observed increases in phytohormone concentrations under XOS treatment are comparable to those induced by the well-established hormone IBAP, underscoring the potential of XOS as an effective agent for promoting root growth.

### 2.3. Transcriptome Analysis 

To explore the molecular basis of root elongation growth stimulated by XOS, we conducted RNA sequencing (RNA-seq) analysis on lettuce roots collected 2 h after treatment, with three biological replicates for each condition. The overall expression profiles of lettuce roots treated with XOS, IBAP, and WT were compared and analysed (Appendix A). Pearson’s correlation coefficients among the three replicates for each treatment exceeded 92%, indicating high reproducibility (Appendix A). Principal component analysis (PCA) clearly distinguished the nine samples into three distinct groups, corresponding to the XOS, IBAP, and WT treatments, respectively (Figure 5A) This segregation indicates significant differences in gene expression profiles between the treatments.

Differential expression analysis revealed that both XOS and IBAP treatments induced substantial transcriptional responses compared to the wild type (WT), highlighting specific pathways associated with root growth and development. IBAP treatment resulted in extensive transcriptomic changes relative to WT, as shown in Figure 5. A total of 3086 differentially expressed genes (DEGs) were identified in the XOS treatment, including 1986 up-regulated genes (log_2_|FC| ≥ 1, padj < 0.05) and 1820 down-regulated genes (Figure 5B). Similarly, IBAP treatment had a pronounced effect on the transcriptome of lettuce roots, resulting in 4554 DEGs, with 2516 up-regulated and 2038 down-regulated genes. Notably, there was a significant overlap in the genes affected by XOS and IBAP treatments, with 62.7% of DEGs induced by XOS also being responding to IBAP (Figure 5C). 

### 2.4. Analysis of GO Pathway Enrichment

To clearly delineate the potential functions of the differentially expressed genes (DEGs) induced by XOS treatment and their associated biological processes, we conducted Gene Ontology (GO) enrichment analyses for both up-regulated and down-regulated genes. The DEGs functionally annotated by GO analysis revealed enrichment in 2966 GO terms in the RNA-seq data of lettuce roots treated with XOS, of which 826 terms were associated with molecular function, 1737 with biological processes, and 403 with cellular components (Appendix A). For instance, DEGs categorised under biological processes were notably related to cellular manganese ion homeostasis, manganese ion transmembrane transporter activity, microtubule motor activity, carbohydrate metabolic process, organic cyclic compound biosynthetic process, and other biological processes, involving a total of 1900 up-regulated and down-regulated genes (Figure 6). In the category of molecular function, DEGs were significantly enriched in manganese ion transmembrane transporter activity, microtubule motor activity, hydrolase activity acting on glycosyl bonds, and microtubule binding. In the cellular component category, DEGs were predominantly enriched in the origin recognition complex. These results suggest that XOS-induced genes regulating lettuce root growth are primarily involved in biological processes and molecular functions. These results suggest that XOS-induced genes regulating lettuce root growth are primarily involved in biological processes and molecular functions.

For IBAP treatment, similar GO analysis identified significant enrichment in 3103 GO terms, with 888 terms associated with molecular function, 1797 with biological processes, and 418 with cellular components (Appendix A). The DEGs in the biological process category for IBAP treatment included responses to cellular manganese ion homeostasis, manganese ion homeostasis, microtubule-based movement, organic cyclic compound biosynthetic process, and oxidation–reduction process. In the molecular function category, significant enrichment was observed in manganese ion transmembrane transporter activity, microtubule motor activity, microtubule binding, oxidoreductase activity, and hydrolase activity hydrolysing O-glycosyl compounds. Within the cellular component category, DEGs were notably enriched in the DNA packaging complex, nucleosome, chromatin, and intracellular non-membrane-bound organelle. Interestingly, the GO enrichment results for both XOS and IBAP treatments showed substantial overlap, particularly in the categories of biological processes and molecular functions. This indicates that many of the pathways through by which XOS and IBAP regulate root growth are shared. Specifically, both treatments prominently featured enrichment in manganese ion homeostasis, microtubule-associated processes, and various metabolic and biosynthetic processes, underscoring the consistency in their modes of action.

### 2.5. Analysis of KEGG Pathway Enrichment

To further elucidate the biochemical and molecular responses, KEGG pathway enrichment analysis was performed to integrate the regulatory pathways of differentially expressed genes (DEGs) in lettuce roots induced by XOS treatment. The top 20 enriched KEGG pathways are presented in Figure 7. Among the highly represented pathways were starch and sucrose metabolism, phenylpropanoid biosynthesis, and plant hormone signal transduction. Notably, DEGs were significantly enriched (q value < 0.05) in the starch and sucrose metabolism pathway, with 26 up-regulated and 14 down-regulated DEGs. This indicates that XOS treatment has a profound effect on carbohydrate metabolism. Additionally, the plant hormone signal transduction pathway, involving 55 DEGs, was also found to play a crucial role in the response to XOS (Appendix A). This pathway primarily included components of auxin (IAA), cytokinin, brassinosteroid (BR), abscisic acid (ABA), and ethylene (ET) signalling, with 19 DEGs up-regulated and 36 DEGs down-regulated. 

Further analysis revealed that XOS treatment enriched several other important pathways, including DNA replication, homologous recombination, mismatch repair, cutin, suberine, and wax biosynthesis, base excision repair, cyanoamino acid metabolism, and fatty acid degradation. The enrichment of pathways such as plant hormone signal transduction and cutin, suberine, and wax biosynthesis in XOS-treated samples indicates unique responses potentially related to enhanced stress tolerance and signalling mechanisms. In comparison, IBAP (control) treatment resulted in the enrichment of pathways related to DNA replication, mismatch repair, starch and sucrose metabolism, homologous recombination, phenylpropanoid biosynthesis, pyrimidine metabolism, nucleotide excision repair, and fatty acid degradation. These pathways represent baseline activities in DNA repair, nucleotide metabolism, and carbohydrate and lipid metabolism.

Interestingly, the KEGG enrichment results for both XOS and IBAP treatments showed substantial overlap, particularly in the categories of starch and sucrose metabolism, phenylpropanoid biosynthesis, and plant hormone signal transduction. This indicates that while XOS induces unique pathways related to stress tolerance, a large proportion of pathways through by which XOS and IBAP regulate root growth are shared. Both treatments prominently featured pathways involved in carbohydrate metabolism, DNA repair, and lipid metabolism, underscoring the shared and unique pathways enriched by both treatments.

### 2.6. Critical Pathway and Functional Genes

Further investigation of the starch and sucrose metabolism pathway showed that the DEGs involved in the pathway were mainly associated with glycosidases, hexosyltransferases, and nucleotidyltransferases (Appendix A), which might be actively participate in the XOS-induced root stimulation of lettuce. In this pathway, we identified six up-regulated and two down-regulated genes in the hexosyltransferase category, while two up-regulated and one down-regulated gene were found in the nucleotidyltransferase category. The glycosidase pathway exhibited more DEGs than others, with ten up-regulated and seven down-regulated genes. Among them, genes encoding beta-glucosidase proteins (bgl BoGH3B-like, bgl12-like, bgl13-like, and bgl24-like genes), beta-fructofuranosidase proteins (soluble isoenzyme I-like, insoluble isoenzyme I-like, and insoluble isoenzyme CWINV3-like genes), and endoglucanase proteins (EP8-like and EP17-like genes) were significantly up-regulated. Conversely, genes encoding beta-glucosidase proteins (BGL40, BGL44-like, BGL46-like, and BoGH3B-like genes) and endoglucanase proteins (EP24-like and EP25-like genes) were remarkably down-regulated.

Additionally, several DEGs enriched in the plant hormone signal transduction pathway, such as those involved in auxin, brassinosteroid, and cytokinin pathways, were also identified. As shown in Appendix A, the auxin pathway had eight down-regulated and nine up-regulated genes, while the brassinosteroid pathway included five down-regulated and four up-regulated genes. In the cytokinin pathway, nine down-regulated genes were detected, with only one up-regulated gene. Notably, genes encoding auxin-responsive proteins (SAUR23-like, SAUR50-like, SAUR63, SAUR71-like, SAUR72-like, and IAA27-like genes), two-component response regulator proteins (ARR3-like, ARR5-like, ARR17-like, OPP9-like, ORR5, and ORR9 genes), probable xyloglucan endotransglucosylase/hydrolase protein 23, and BRI1 kinase inhibitor 1-like protein were significantly down-regulated (Figure 8). In contrast, genes encoding auxin transporter-like2 protein, putative indole-3-acetic acid-amido synthetase GH3.9 protein, auxin-induced protein 6B protein, probable indole-3-acetic acid-amido synthetase GH3.1 protein, auxin-responsive protein SAUR63, auxin-responsive protein IAA16-like protein, and brassinosteroid insensitive1-associated receptor kinase1 were notably up-regulated. In addition, genes involved in nitrogen metabolism (glutamine synthetase, both chloroplastic and non-chloroplastic forms), high-affinity nitrate transport (2.1-like, 2.4-like, and 2.6-like), and carbon metabolism (beta carbonic anhydrase 5 and carbonic anhydrase 2-like) were markedly increased (Appendix A). These results indicate that XOS enhances key metabolic pathways and nutrient uptake mechanisms, supporting robust root development.

These findings highlight the intricate molecular mechanisms through by which XOS influences root growth in lettuce, emphasising the significant roles of glycosidases, hexosyltransferases, and nucleotidyltransferases in carbohydrate metabolism, as well as the involvement of various plant hormone signalling pathways. The integration of these pathways underscores the complex regulatory networks modulated by XOS treatment, contributing to enhanced root development and growth responses.

### 2.7. Results of RT-PCR Confirmation

To validate the transcriptomic sequencing results, quantitative real-time PCR (RT-PCR) was utilized to assess the expression levels of differentially expressed genes (DEGs) induced by xylooligosaccharide (XOS) treatment in lettuce (Figure 9). Thirteen DEGs were randomly selected for analysis, including auxin-responsive protein SAUR72-like, hexokinase-2, serine/threonine-protein kinase SRK2E-like, and several two-component response regulators. The correlation coefficients (R) of log2-transformed fold change values between RT-PCR and RNA-seq were 0.769 and 0.841 for the IBAP and XOS treatments, respectively. The RT-PCR results largely corroborated the expression profiles obtained from the RNA sequencing analysis, confirming the reliability of the transcriptomic data. However, it was observed that for several genes (two-component response regulator ARR5-like, two-component response regulator ARR5, two-component response regulator ARR3-like, two-component response regulator ORR9-like, and auxin-induced protein 22D-like), the relative expression trends in the IBAP treatment differed between RNA-seq and RT-PCR results. Such differences may arise from several factors, including technical variations between the two methodologies, the dynamic range of gene expression detection, or biological variations such as post-transcriptional modifications or differential mRNA stability. Despite these differences, the overall correlation coefficients indicate a strong agreement between the RNA-seq and RT-PCR data, supporting the robustness of the transcriptomic analysis.

## 3. Discussion 

### 3.1. Mechanistic Insights into XOS-Mediated Root Development and Gene Expression Alterations

The root system is the lifeline of a plant, crucial for water and nutrient uptake and integral to its response to environmental challenges [31,32,33,34]. A well-developed root system not only enhances crop productivity but also improves quality, resilience, and overall plant health [35,36]. Exploring the application of external agents to promote root development has emerged as a promising agronomic strategy [37,38,39]. Such interventions can lead to significant improvements in agricultural yields and sustainability, while also potentially reducing the need for chemical fertilizers and enhancing soil health [36]. Among the various substances studied, oligosaccharides have attracted significant interest for their potential to enhance root growth in plants [20,40,41,42]. Despite extensive research showing their effectiveness, the precise mechanisms through which oligosaccharides, especially xylooligosaccharides (XOSs), influence root development remain largely elusive. This study combines phenotypic, physiological, and transcriptomic analyses to elucidate the regulatory effects of XOS on gene expression and root development in lettuce. Our results suggest that XOS enhances root development by modulating key pathways involved in energy production, hormone regulation, and nitrogen metabolism (Appendix A). XOS treatment significantly enhances the biomass, total root length, total volume, and total surface area in lettuce roots, corroborating previous studies (Figure 1 and Table 1). Notably, XOS exerted a pronounced positive effect on both adventitious and lateral roots, with increases of 28.55% and 29.86%, respectively (Figure 2). However, no significant difference was observed in the ratio of adventitious to lateral root length across different treatments, suggesting that XOS promotes root growth primarily by stimulating the formation and elongation of adventitious roots.

Over the years, scholars have explored the function and mechanisms of oligosaccharides in regulating plant growth, focusing on phenotypic characteristics and molecular responses [43,44]. To better understand the molecular response of lettuce roots to XOS, we employed RNA-seq analysis. Our results identified 1986 up-regulated and 1821 down-regulated DEGs in lettuce roots under XOS treatment (Figure 5). KEGG enrichment analysis indicated significant enrichment in starch and sucrose metabolism pathways among the top-20 pathways (Figure 7). Carbohydrates, as primary products of photosynthesis, provide energy and metabolic intermediates essential for plant growth [45,46]. Our study demonstrated that exogenous XOS signals activate genes related to endogenous carbohydrate metabolism. DEGs encoding beta-glucosidase and beta-fructofuranosidase proteins, which are crucial for cell wall metabolism, hormone activation, and stress responses, were notably enriched (Appendix A) [47]. These findings suggest that XOS-induced root promotion is likely related to accelerated sugar metabolism, which supplies more energy and intermediates for root development and hormone synthesis.

Phytohormone signalling, including auxin, cytokinin, and brassinosteroids, plays a pivotal role in various processes of plant growth and development, such as macromolecule biosynthesis, photosynthesis, and environmental responses [48,49,50]. Relevant studies have indicated that oligosaccharide-induced root development is likely mediated through auxin pathways [51,52]. For instance, alginate oligosaccharides (AOSs) applied to rice seedlings significantly increased IAA concentrations, correlating with root formation and elongation, while the promotion effect was inhibited by the auxin transport inhibitor TIBA [21,53]. To verify the role of phytohormones in response to exogenous XOS, we analysed related plant hormones in XOS-treated lettuce. Our results showed a substantial increase in IAA, BR, and ZR levels, suggesting that XOS drives adventitious root growth by enhancing phytohormone-mediated signalling pathways (Figure 4).

Furthermore, our analysis revealed that DEGs induced by XOS were significantly enriched in plant hormone signalling pathways, particularly the auxin signalling pathway (Figure 8). RNA-seq data indicated that several ARF-related genes were repressed by XOS, while genes encoding AUX/IAA proteins, important negative regulators of auxin signalling, were down-regulated. This suggests that XOS enhances ARF transcriptional activity by inhibiting AUX/IAA gene expression. Additionally, genes encoding GH3 proteins, which conjugate excess IAA to amino acids to maintain hormonal homeostasis, were up-regulated [54]. This indicates that XOS maintains IAA balance in roots, contributing to increased root biomass. Genes encoding the auxin influx carrier AUX1, which is involved in auxin transport and root development, were also up-regulated, further supporting the role of auxin signalling in XOS-induced root growth.

In addition, genes involved in the tryptophan-dependent auxin biosynthesis pathway were activated by XOS, including YUCCA10 and YUCCA5, accounting for the increased IAA content in XOS-treated roots. Appropriate increases in free IAA content not only regulate growth and development but also enhance nutrient absorption and distribution [55]. Our study found that transcripts encoding high-affinity nitrate transporters (NRT2.1-like, NRT2.4-like, and NRT2.6-like) were up-regulated, suggesting that XOS-induced nitrate uptake contributes to root promotion (Appendix A). Interestingly, genes controlling type-A ARRs, negative regulators of cytokinin signalling, were highly down-regulated, while type-B ARR activity, which promotes cytokinin signalling, was enhanced (Appendix A). This indicates that XOS may stimulate cell division and adventitious root growth through auxin-cytokinin antagonism. These findings provide a comprehensive understanding of the molecular mechanisms underlying XOS-induced root growth in lettuce, highlighting the significant roles of sugar metabolism and phytohormone signalling. 

### 3.2. Comparative Analysis of XOS and IBAP: Divergent Pathways to Root Enhancement

Comparative analysis between XOS and the traditional plant hormone indole-3-butyric acid potassium salt (IBAP) reveals both shared and distinct mechanisms driving root enhancement. Both XOS and IBAP significantly promote root growth and increase root biomass, total root length, volume, and surface area (Figure 1 and Table 1). However, their underlying mechanisms of action exhibit notable differences. While both treatments elevated IAA levels, the increase was more pronounced in IBAP-treated plants, suggesting that IBAP might be more potent in directly boosting auxin levels (Figure 4) [56]. In contrast, XOS uniquely increased levels of other phytohormones, such as ZR and JA-ME, indicating a broader hormonal modulation. This broader hormonal impact suggests that XOS may facilitate root growth through a more integrated network of growth regulators, potentially enhancing plant resilience and adaptability to various environmental conditions.

Transcriptomic data revealed substantial transcriptional responses to both XOS and IBAP, but with different emphases. XOS treatment led to significant up-regulation of genes involved in carbohydrate metabolism, particularly those related to starch and sucrose metabolism (Figure 7). This indicates that XOS enhances root growth by improving carbohydrate availability and metabolism, providing the essential energy and building blocks for root development. In contrast, IBAP treatment showed a stronger effect on genes directly involved in auxin signalling and transport, highlighting its more targeted action on the auxin pathway (Figure 8) [57]. Additionally, the differential gene expression patterns suggest that XOS influences pathways related to stress response and nutrient uptake (Appendix A). For instance, genes associated with nitrate uptake were up-regulated under XOS treatment, indicating improved nutrient assimilation capabilities. This broader impact on various physiological processes positions XOS as a versatile and sustainable option for agricultural practices, potentially offering advantages over traditional plant hormones like IBAP.

## 4. Materials and Methods

### 4.1. Materials

Unless otherwise stated, the highly purified (purity exceeds 85%) xylooligosaccharide (molecular formula, (C_5_H_11_O_6_(C_6_H_10_O_5_)_4_); molecular weight, 827) was collected from Showa Denko K.K. (Tokyo Metropolis, Japan). The oligomer was composed of beta-1,4-linked xylose and exhibited fairly good hydrophilic properties. The oligomers predominantly exist as tetramers and pentamers. Lettuce seeds (Lactuca sativa var. ramosa Hort.) were sourced from China Vegetable Seed Technology Co., Ltd. (Beijing, China).

### 4.2. Plant Culture and Experimental Design

Lettuce (Lactuca sativa var. ramosa Hort.) was cultivated under controlled conditions in a plant factory, with 60% relative humidity and temperatures maintained at 25 ± 1 °C during the day and 23 ± 1 °C at night, on a 12 h/12 h cycle. Illumination was provided at 200 µmol·m^−2^·s^−1^ using white LED lights. Seeds were surface-sterilized using a 10% sodium hypochlorite solution for 10 min and rinsed thoroughly with distilled water. Germination occurred on seeding-raising plates (100 × 40 × 12 cm) under white LED lights for two weeks, ensuring uniform germination and healthy root development. At the third true leaf stage (16-day-old seedlings), seedlings were transferred to hydroponic polyethylene (PE) pots (45 × 45 × 12 cm) and exposed to red and blue LED lighting at a ratio of 4R:1B. An adjusted Hoagland solution with a pH of 6.0 was used, containing the following components (mM): 0.75 K_2_SO_4_, 0.5 KH_2_PO_4_, 0.1 KCl, 0.65 MgSO_4_·7H_2_O, 0.1 EDTA-Fe, 1.0 × 10^−3^ H_3_BO_3_, 1.0 × 10^−3^ MnSO_4_·4H_2_O, 1 × 10^−3^ ZnSO_4_·7H_2_O, 1 × 10^−4^ CuSO4·5H_2_O, 5 × 10^−6^ H_2_MoO_4_, and 5.0 Ca(NO_3_)_2_·4H_2_O (N10). Two days post-transplantation, three treatments were initiated: Tr1—sterile distilled water (control), Tr2—20 mg L^−1^ indole-3-butyric acid potassium salt (IBAP), and Tr3—40 mg L^−1^ xylooligosaccharide (XOS). Treatments were applied through foliar spraying four times at three-day intervals over a ten-day period, using IBAP as a benchmark to compare regulatory effects.

### 4.3. Root Morphology and Phytohormone Measurement

Two days after the final treatment, on the 26th day, six independent biological replicates were collected for each treatment group. Root tissues were separated to assess fresh and dry weights. The morphology of the roots was quantified using the WinRHIZO root analyser system (Regent Instruments, Boul Wilfrid-Hamel, Quebec, QC, Canada). For hormonal analysis, root tissues from the same plants were collected, immediately frozen in liquid nitrogen, and stored at −80 °C until analysis. Phytohormone content of IAA, BR, JA-ME, BR in the root tissue were measured with the Plant Methyl Jasmonate JA-ME (SP39137, spbio), Plant Indoleacetic Acid ELISA kit (SP29769, spbio), Plant Brassinosteroids BR ELISA kit (SP29797, spbio), and Plant Zeatin Riboside ZR ELISA kit (SP29802, spbio) following the manufacturer’s protocols.

### 4.4. Transcriptome Analysis

Total RNA was extracted from treated and untreated lettuce roots using the TRIzol method (Tiangen Biotech, Haidian District, Beijing, China). RNA concentration was quantified using an Agilent 2100 Bioanalyser (Agilent Technologies, Santa Clara, CA, USA), and its purity was assessed with a NanoDrop spectrophotometer (IMPLEN, Munich, Germany). RNA integrity was monitored via electrophoresis on 1% agarose gels to check for contamination and degradation. Sequencing libraries were prepared using the NEBNext^®^ Ultra™ RNA Library Prep Kit for Illumina^®^ (NEB, Ipswich, MA, USA) according to the manufacturer’s instructions, with index codes assigned to each sample to track sequence origins. The PCR products were then purified using the AMPure XP system, and library quality was evaluated using the Agilent Bioanalyser 2100 system. Libraries were sequenced on an Illumina HiSeq 4000 platform by Beijing Allwegene Technology Company Limited (Beijing, China), producing paired-end reads of 150 bp. Raw reads were filtered using an in-house Perl script to obtain high-quality clean reads, with metrics such as Q20, Q30, and GC content of the clean data being calculated. All subsequent analyses were conducted using these high-quality clean reads.

### 4.5. Analysis of Differentially Expressed Genes (DEGs) and Heat-Map Generation

Gene expression levels were estimated using read count values and fragments per kilobase of transcript per million mapped fragments (FPKM). DEGs between the control (WT) group and treatment group were identified based on an adjusted false discovery rate (FDR) with *p*-values < 0.05 and a log_2_ |fold change| ≥ 1. Gene functions were annotated using the Gene Ontology (GO) and Kyoto Encyclopedia of Genes and Genomes (KEGG) databases. DEGs were clustered and analysed for enrichment in GO terms and KEGG pathways using the GOseq R package, which utilizes a Wallenius non-central hyper-geometric distribution, and KOBAS software (2.0). Heat-maps were generated to visualize expression patterns of DEGs related to starch and sucrose metabolism and plant hormone signal transduction pathways. These heat-maps were created using log_2_-transformed (FPKM+1) values in R Studio software 2.8.2.

### 4.6. RT-PCR Confirmation

To confirm the expression levels of target genes, quantitative real-time PCR was conducted. DNA was extracted and reverse-transcribed into cDNA using the PrimeScript™ RT reagent Kit with gDNA Eraser (Thermo Fisher Scientific, Waltham, MA, USA). The subsequent qRT-PCR was performed using TB Green^®^ Premix Ex Taq™ II (Tli RNaseH Plus) (Takara Bio, San Jose, CA, USA), employing primers detailed in Appendix A. For normalization, the gene LsPP2AA3 (Protein phosphatase 2A regulatory subunit A3) served as the reference gene, following the methodology endorsed by Sgamma et al. [58]. Relative quantification of gene expression was achieved through the 2^−ΔΔCT^ approach, as outlined by Livak & Schmittgen [59].

### 4.7. Statistical Analysis

The data presented in the graphs were analysed using SPSS 21 software (IBM Corp., Armonk, NY, USA) and are expressed as mean ± SD values. Statistical differences between treatments and controls were assessed using a one-way ANOVA for multiple comparisons and a two-tailed unpaired Student’s *t*-test for comparisons between two groups. A *p* value of less than 0.05 was considered statistically significant. Graphs were created using Origin 9.0.

## Figures and Tables

**Figure 1 plants-13-01699-f001:**
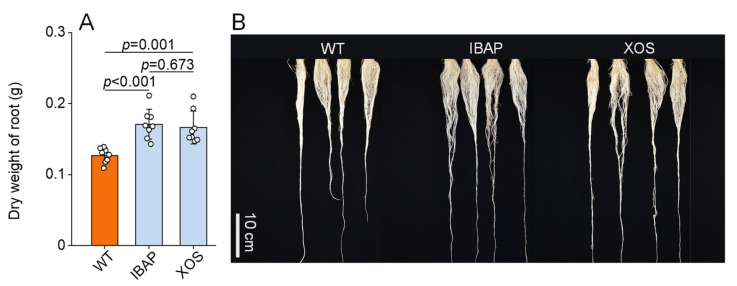
Dry weight (**A**) and phenotypic changes (**B**) of roots in 28-day-old lettuce plants, with and without treatment. The white dot represents the date point. The data represent means ± SD (*n* = 8). *p* values were derived by two-tailed Student’s *t*-test. XOS: foliar application of xylooligosaccharide, IBAP: foliar application of indole-3-butyric acid potassium salt, WT: foliar application of distilled water.

**Figure 2 plants-13-01699-f002:**
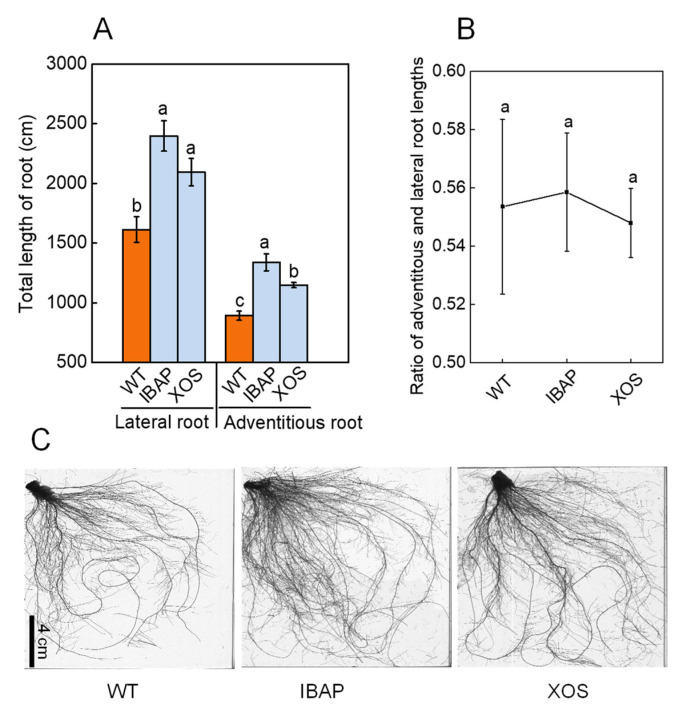
Changes in root length of adventitious roots and lateral roots in lettuce (**A**), the ratio between them (**B**), and rhizotron root images (**C**) of lettuce plants at 28-days old under different treatments. The lateral roots were classified using the WinRHIZO root image analysis system based on the average diameter (lateral roots: 0 < AvgDiam ≤ 0.25 mm; adventitious roots: AvgDiam > 0.25 mm). Results marked with different lowercase letters indicate significant differences (*p* < 0.05) between groups, determined by multiple comparison tests. Each value represents the mean ± SD (*n* = 8).

**Figure 3 plants-13-01699-f003:**
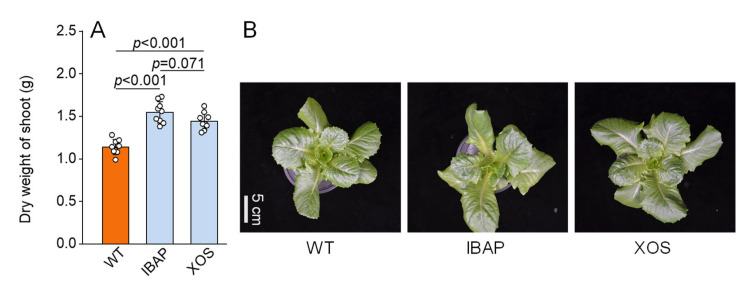
Dry weight (**A**) and phenotypic changes (**B**) of shoots in 28-day-old lettuce plants, with and without treatment. The data represent means ± SD (*n* = 8). *p* values were derived by two-tailed Student’s *t*-test. XOS: foliar application of xylooligosaccharide, IBAP: foliar application of in-dole-3-butyric acid potassium salt, WT: foliar application of distilled water. The white dot represents the date point.

**Figure 4 plants-13-01699-f004:**
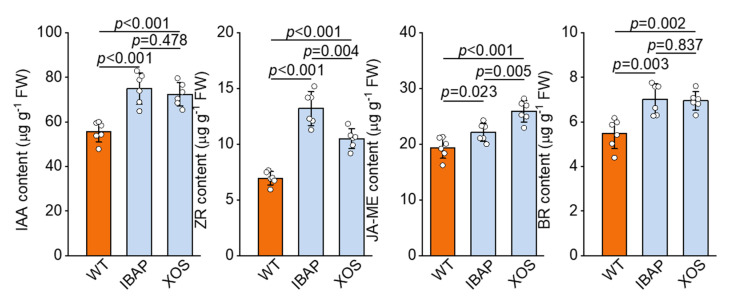
Effect of xylooligosaccharide (XOS) and indole-3-butyric acid potassium salt (IBAP) on phytohormone content in lettuce root. IAA: indole-3-acetic acid; ZR: zeatin riboside; JA-ME: methyl jasmonate; BR: brassinolide. Each value represents the mean ± SD (*n* = 6). *p* values were derived by two-tailed Student’s *t*-test. The white dot represents the date point.

**Figure 5 plants-13-01699-f005:**
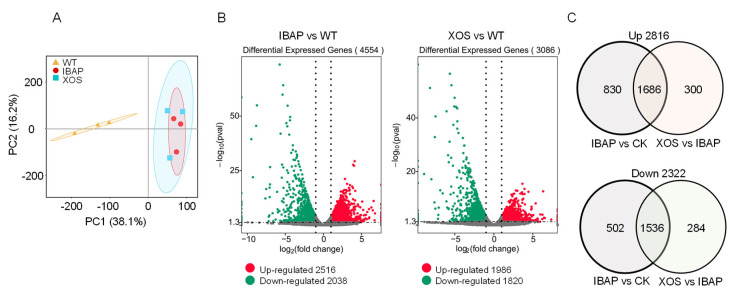
(**A**) Principal component analysis (PCA) of RNA-seq data from lettuce roots with or without treatment. (**B**) Volcano plot analysis of differentially expressed genes (DEGs) in lettuce leaves treated with xylooligosaccharide (XOS) and indole-3-butyric acid potassium salt (IBAP). DEGs with a two-sided adjusted *p* value less than 0.05, corrected using the Benjamini and Hochberg false discovery rate (FDR) method, are considered statistically significant. (**C**) Venn diagram illustrating the overlap between transcriptome datasets from XOS-treated and IBAP-treated lettuce leaves (likelihood ratio test; *p* < 0.05).

**Figure 6 plants-13-01699-f006:**
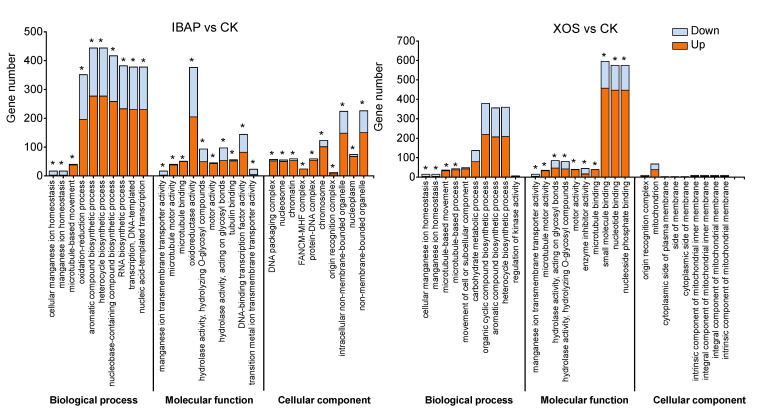
Gene Ontology (GO) classification of differentially expressed genes (DEGs) induced by xylooligosaccharide (XOS) and indole-3-butyric acid potassium salt (IBAP) in lettuce roots. The DEGs are categorized into molecular function, biological process, and cellular component categories. Asterisks denote statistically significant enrichment of terms.

**Figure 7 plants-13-01699-f007:**
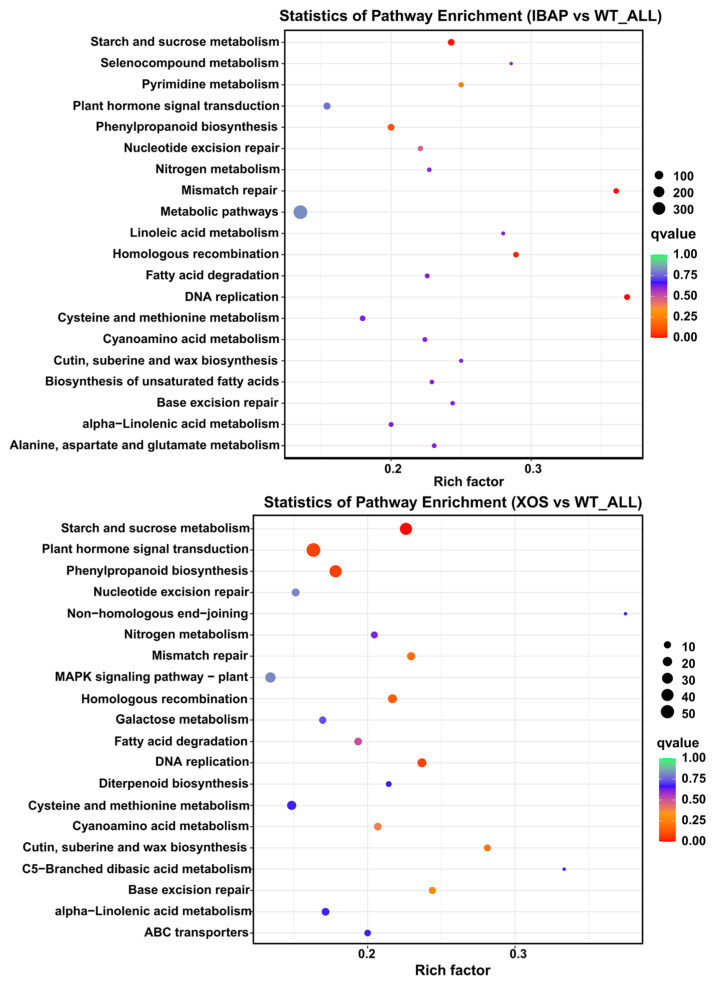
The top 20 up- and down-regulated differentially expressed gene (DEG)-enriched KEGG terms in lettuce roots treated with xylooligosaccharide (XOS) and indole-3-butyric acid potassium salt (IBAP).

**Figure 8 plants-13-01699-f008:**
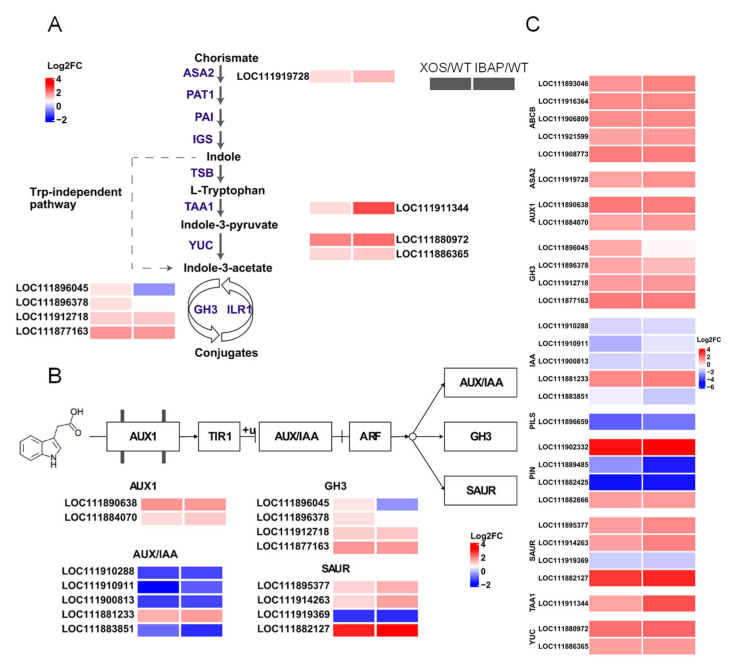
Heatmaps illustrating the expression profiles of key differentially expressed genes (DEGs) related to auxin signalling pathways in response to xylooligosaccharide (XOS) and indole-3-butyric acid potassium salt (IBAP) treatments. Red indicates up-regulation, while blue indicates down-regulation. (**A**) Auxin biosynthesis pathway, (**B**) auxin signal transduction pathway, (**C**) key transcription factors.

**Figure 9 plants-13-01699-f009:**
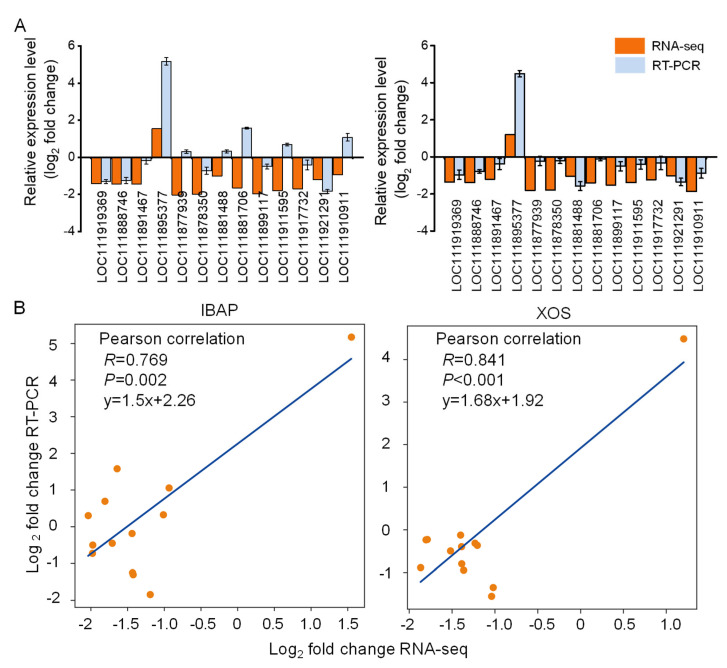
(**A**) Comparison of mRNA expression changes (log_2_ fold change) for 13 differentially expressed genes between RT-PCR and RNA-seq in XOS and IBAP treatment groups versus the control. (**B**) Significant Pearson correlation (*p* < 0.05) between fold changes in gene expression by RTPCR and RNA-seq in IBAP and XOS treatments.

**Table 1 plants-13-01699-t001:** The main characteristics of lettuce roots.

Treatment	Material	Length (cm)	Volume (cm^3^)	Surface Area (cm^2^)
WT	water	2508.09 ± 134.67 b	1.90 ± 0.03 b	244.88 ± 8.21 b
IBAP	indole-3-butyric acid potassium saltin	3737.72 ± 194.50 a	2.59 ± 0.22 a	348.79 ± 24.12 a
XOS	xylooligosaccharide	3245.52 ± 137.27 a	2.31 ± 0.08 a	307.18 ± 11.75 a

Distinct lowercase letters denote significant differences among treatments at the 0.05 significance level. Each value is presented as the mean ± SD (*n* = 8).

## Data Availability

The RNA-seq data have been deposited in the Sequence Read Archive (SRA) of the National Center for Biotechnology Information (NCBI) under the accession number PRJNA824284. All data supporting the conclusions of this study are included within the main manuscript and the Appendix A.

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
