# Peer review of "Xylooligosaccharides Enhance Lettuce Root Morphogenesis and Growth Dynamics"

_plants, 2024, doi:10.3390/plants13121699_

Round 1
Reviewer 1 Report
Comments and Suggestions for Authors
Root development is crucial for plant growth. This manuscript explores the impact of xylooligosaccharide (XOS) treatment on root development, hormone fluctuations, and gene expression in 28-day-old lettuce. The findings confirm that exogenous XOS treatment significantly enhances lettuce root growth through enhanced carbohydrate metabolism and broad hormonal modulation. This study is systematic and comprehensive, and the manuscript is well written, resulting in correct conclusions. Here are some suggestions/questions:
1. Provide details on hormone assay kits, such as number, manufacturer, etc;
2. Indicate the date of using the database to compare and identify DEGs, as the content in the database is continuously increasing;
3. What is the basis for the concentration of XOS and IBAP used in the manuscript?
4. Both treatments in the manuscript analyzed 20 DEG enriched KEGG pathways, but only 8 pathways in each treatment had a p-value < 0.05;
5. The multiple of changes in DEGs should be presented as mean±SD;
6. It is recommended to make table S8 a figure and place it in the manuscript;
7. The author needs to rewrite section 2.7. Currently, there has been no analysis or discussion of the results in table S8, such as the expression of two-component response regulator ARR17-like under XOS treatment showed completely opposite trends in RNA-Seq and qRT-PCR. At the same time, it is recommended that the author conduct correlation analysis on the data in RNA-Seq and qRT-PCR, and calculate the R value;
8. Line 321, ten selected DEGs? Are the specific primers for these selected DEGs designed by the author or referenced from literature?
Reviewer 2 Report
Comments and Suggestions for Authors
1. The order of affiliations for author Jiqing Song should be corrected from “1,3 to 1,2”.
2. Lines 97-99: IBAP is a well-established plant hormone. To compare the effects of exogenous XOS with those of a known hormone on root growth, please cite relevant literature to support this comparison.
3. Please include images in Figure 2 that compare the morphology of lateral and adventitious roots after XOS and WT treatments.
4. In Figure 3B, the growth stages of lettuce treated with IBAP are not consistent with those of WT and XOS treatments. Please check to ensure that plants from different treatments were selected at the same time points. Additionally, the images lack a scale bar.
5. There are errors in the differential gene expression data presented in Figure 5C. Please carefully review and correct these discrepancies.
6. In previous experiments, the effects of XOS on root and endogenous hormones were assessed after 2 days of treatment. Why was a 2-hour treatment time chosen for the transcriptome analysis? Is there a specific rationale for this selection?
7. Lines 201-205: The oxidoreductase activity is the most enriched category, but the authors have neglected this point. Please carefully review and address this issue.
8. Lines 318-328: The qPCR verification should be presented in graphical form within the main text. What is the basis for selecting the reference gene? Additionally, it is suggested that the authors use two other reference genes for the qPCR analysis, such as SAND and TUBULIN.
Comments on the Quality of English LanguageThe overall quality of English language in the manuscript is satisfactory
Reviewer 3 Report
Comments and Suggestions for Authors
The article titled “Xylooligosaccharides Enhance Lettuce Root Morphogenesis and Growth Dynamics” presents the regulatory effects of xylooligosaccharides (XOS) on lettuce root growth, comparing their impact with that of indole-3-butyric acid potassium salt (IBAP). The study showed that treatment with XOS led to a substantial increase in root dry weight (30.77%), total root length (29.40%), volume (21.58%), and surface area (25.44%) compared to the water-treated control which is on par with those induced by IBAP. Moreover, the phytohormone profiling disclosed marked increases in indole-3-acetic acid (IAA), zeatin riboside (ZR), methyl jasmonate (JA-ME), and brassinosteroids (BRs) following XOS application. Furthermore, authors used RNA-seq technology to investigate the effects of XOS on gene expression in lettuce roots, with IBAP serving as a comparative control to elucidate the differences and similarities in their regulatory mechanisms. The study identified 3,807 DEGs in the roots of XOS-treated plants, which were enriched in pathways associated with manganese ion homeostasis, microtubule motor activity, and carbohydrate metabolism. Almost 62.7% of the DEGs responsive to XOS also responded to IBAP, underscoring common regulatory mechanisms. Using different physiological, morphological, and biochemical analyses the study shows XOS significantly promotes lettuce root growth through a multifaceted mechanism involving enhanced carbohydrate metabolism and broad hormonal modulation. While both XOS and IBAP effectively enhance root development, XOS offers a more integrated approach, influencing a wider range of physiological processes according to the study. By revealing the complex network of metabolic and signaling pathways involved in XOS- and IBAP-mediated root growth regulation, this work provides valuable insights for the application of these substances in agriculture. The manuscript will be useful for researchers who are interested in root development, and adaptations and understand their effect on plant physiology and metabolism and crop production. Using different effective tools for improving crop productivity and resilience, offers significant advantages over traditional plant hormones. In my opinion, the manuscript is suitable for publication in Plants Journal. The manuscript needs some English corrections, including mistakes, spelling, articles, etc.
Comments on the Quality of English LanguageMinor editing of the English language is required.
Round 2
Reviewer 2 Report
Comments and Suggestions for Authors
I appreciate the revisions and the effort the authors have made. However, I still have concerns about the qPCR results. These results need to be included in the main Figure and should be based on at least three biological replicates. Additionally, I recommend using multiple reference genes for verification. The relative expression levels should be presented in graphical form. This information needs to be clearly detailed throughout the paper.
Round 3
Reviewer 2 Report
Comments and Suggestions for Authors
The authors have enhanced the manuscript from the original preprint. The conclusions are now well supported by the evidence presented.